# Food Selectivity in Children with Autism: Guidelines for Assessment and Clinical Interventions

**DOI:** 10.3390/ijerph20065092

**Published:** 2023-03-14

**Authors:** Marco Esposito, Paolo Mirizzi, Roberta Fadda, Chiara Pirollo, Orlando Ricciardi, Monica Mazza, Marco Valenti

**Affiliations:** 1Autism Research and Treatment Centre Una Breccia Nel Muro, 00168 Rome, Italy; 2Department of Applied Clinical Sciences and Biotechnology, University of L’Aquila, 67100 L’Aquila, Italy; 3Department of Education, Psychology, Communication, University of Bari, 70121 Bari, Italy; 4Department of Pedagogy, Psychology, Philosophy, University of Cagliari, 09100 Cagliari, Italy; 5Regional Centre for Autism, Abruzzo Region Health System, 67100 L’Aquila, Italy

**Keywords:** food selectivity, autism spectrum disorder, applied behavior analysis, sensory processing, parent training

## Abstract

Autisms Spectrum Disorders (ASD) are characterized by core symptoms (social communication and restricted and repetitive behaviors) and related comorbidities, including sensory anomalies, feeding issues, and challenging behaviors. Children with ASD experience significantly more feeding problems than their peers. In fact, parents and clinicians have to manage daily the burden of various dysfunctional behaviors of children at mealtimes (food refusal, limited variety of food, single food intake, or liquid diet). These dysfunctional behaviors at mealtime depend on different factors that are either medical/sensorial or behavioral. Consequently, a correct assessment is necessary in order to program an effective clinical intervention. The aim of this study is to provide clinicians with a guideline regarding food selectivity concerning possible explanations of the phenomenon, along with a direct/indirect assessment gathering detailed and useful information about target feeding behaviors. Finally, a description of evidence-based sensorial and behavioral strategies useful also for parent-mediated intervention is reported addressing food selectivity in children with ASD.

## 1. Introduction

Children with Autism Spectrum Disorders (ASD) are characterized by social communication deficit and a tendency to engage in a pattern of restricted and repetitive behaviors including commonly associated comorbidities such as language disorders, hyperactivity, anxiety, challenging behaviors, food selectivity, and sensory [1]. Exploring feeding issues in ASD, we can observe either medical or sensorial/behavioral characteristics which influence food refusal or restricted food preferences of autistic children during mealtimes. During mealtimes, young children explore food with sense organs, gradually acquiring more self-knowledge through taste, touch, and smell perceptions [2]; these experiences comprising imitation and reciprocal exchanges are also characterized by support, affection, and fun. At the same time, parental feeding practices including families’ food preferences could influence children’s eating behaviors by modeling the intake of fruit and vegetables, limiting snacks, allowing a wide variety of food, preparing specific meals as different from those of the rest of the family [3]. Also, family meals allow the development of the social components of nutrition, as starting from them the child shows the ability to imitate the nutritional choices, patterns, and behaviors of family members [4]. Moreover, the first year of life should be considered a critical period since children consume a single kind of food (breast milk or concentrate). Successively, they gradually assume a variety of foods included in the parents’ diet [5]. In fact, during the period of early development, eating behaviors cease to be merely determined by biological factors since they are influenced by social and contextual aspects such as parental practices and/or observation of peer behavior [6]. Hence, weaning is described as the transactional period that starts from 4 months until the end of the second year of life and marks the transition to the assumption of more solid consistencies [7]. Some studies have confirmed that family behavior normally permits a functional development of feeding in children; however, it could result also in an important component in the development and/or maintenance of eating problems [8]. Parent–child interactions could involuntarily maintain a restriction on children’s diet or facilitate less exposure to a variety of foods [9]. Consequently, choosiness displayed by children during mealtimes could induce other challenging behaviors that reduce food and nutritional intake. Likewise, a meta-analysis [10] has suggested that families can reduce the feeding issues of children (overweight, intake of junk food, and other eating disorders) increasing the consumption of healthy foods. Generally, regardless of the presence of neuropsychiatric disorders, children can exhibit dysfunctional feeding behaviors aimed at avoiding the consumption of certain foods, for example, the dysfunctional behaviors that can occur during the mealtimes including screams, crying, irritability, hetero-direct, and self-directed aggression, escape (moving away from the chair), distress reaction, turning the head to the other side, chewing without swallowing, spitting out and vomiting. Even if the diverse challenging behavior of children displayed at mealtime could cause distress in families’ routines, in the last decades, many researchers have investigated the influence of feeding issues on the nutritional intake of children. Dysfunctional feeding behaviors might be particularly critical for autistic children, who are more likely to avoid food compared to typically developing children due to their sensory issues and their restricted interests and behaviors.

The current narrative review aims to describe the main topics surrounding the phenomenon of food selectivity in children with ASD, in terms of diagnosis, medical and psychological theories, clinical assessments, and interventions with evidence.

The following chapter two describes the impact of food selectivity in children with ASD, regarding the behavioral aspects of diagnosis, the possible nutritional impairments, unhealthy weight, and the hypotheses about the nature of the phenomenon and its impact on parental stress. Chapter three includes the direct and indirect strategies for assessing food selectivity and its comorbidities, and chapter four describes medical, sensorial, and behavioral interventions which have gathered empirical evidence. Chapter five discusses the results of the research studies, and finally, we have included the conclusions.

## 2. Diagnosis and Prognosis of Food Selectivity in Autism

In the new edition of DSM-5 (https://www.ncbi.nlm.nih.gov/books/NBK519712/table/ch3.t18/, accessed on 9 February 2023), nutrition and eating disorders during childhood include pica, avoidant/restrictive disorder of food intake, rumination disorder, along with eating disorders. Therefore, a persistent eating disorder results in a reduced consumption of food and influences physical health or psychosocial functioning. For example, avoidant/restrictive food intake disorder (ARFID) is characterized by avoidance or restriction of food intake preventing the necessary requirements for nutrition or daily caloric intake. The scientific literature considers this diagnostic category an alternative expression of food selectivity [11,12]. Consequently, in clinical practice, it is important to distinguish different aspects of food refusal before the implementation of an effective treatment. Firstly, clinicians have to examine if the feeding problem reports an organic nature or a behavioral one. For example, symptoms displayed by children such as vomiting and some challenging behaviors could be associated merely with bio-medical factors such as gastrointestinal reflux and significant deficits in nutritional intake [13]. Conversely, a severe symptom of food refusal could be characterized by dysfunctional behaviors that have the function to obtain social attention or escape [14]. Naturally, both medical and behavioral dimensions could coexist in the same developmental stage.

Specifically, regarding food selectivity in children with ASD, some authors have shown that children with autism reject more food (accepting low-consistency food as puree) than typically developing children (TDC) [15,16]. Furthermore, children with ASD consume less fruit, dairy products, vegetables, proteins, and starch than children without a diagnosis [17]. Likewise, the results of a study carried out with children aged three to five showed that children with ASD, with respect to controls, preferred foods of a certain consistency (68% vs. 5%), are choosier about food (79% vs. 16%), more reticent to try new foods (95% vs. 47%) and assumed a restricted variety of food (58% vs. 16%) [18]. In the last decades, a main study conducted by Bandini, Anderson, Curtin, Cermak, Evans, et al. [19] defining food selectivity as food refusal, restricted variety of food, and single-food intake, compared children with ASD and TDC, evaluating the impact on the related nutritional intake. This study, based on the Food Frequency Questionnaire (FFQ), showed a significantly greater refusal of food (especially vegetables) of children with ASD compared to TDC. During the last ten years, the studies on food selectivity in children with ASD have increasingly shed light on various dimensions of the problem. A parent report on food selectivity study [20], examined 525 children aged 2–18 years with and without atypical development (ASD, PDD-NOS, and Asperger’s disorder). Individuals with an ASD were reported to have significantly more food selectivity than both the atypically developing group and the TDC. In addition, food refusal showed a decrease across childhood, especially in the Asperger’s disorder group. In fact, a significant study examined the food selectivity in children with ASD longitudinally [21]. A total of 52 parents of children with autism were surveyed 20 months after completing an initial questionnaire. First and second surveys each contained identical parent-response items to categorize food selectivity levels and a scale to measure sensory over-responsivity. A new scale was added at time two to measure restricted and repetitive behaviors. Results comparing time one to time two indicated no change in food selectivity level and a stable, significant relationship between food selectivity and sensory over-responsivity. These results support the chronicity of food selectivity in young children with autism and the consistent relationship between food selectivity and sensory over-responsivity. Hence, some studies investigate if the food selectivity of children persists into adolescence. In one more recent study, food selectivity was evaluated in 18 children with ASD at two time points (mean age = 6.8 and 13.2 years). While food refusal improved overall, the authors did not observe an increase in food repertoire (number of unique foods eaten). These findings support the need for interventions early in childhood to increase the variety and promote healthy eating among children with ASD [22]. Another interesting study aimed at examining the mealtime behaviors and food preferences of adolescents with ASD [23]. An online questionnaire on mealtime behavior and food preferences of ASD students was conducted by caregivers including parents, and the average age of ASD students was 14.1 ± 6.1. The analysis of mealtime behavior resulted in a classification into three clusters: cluster one, the “low-level problematic mealtime behavior group”; cluster two, the “mid-level problematic mealtime behavior group”; and cluster three, the “high-level problematic mealtime behavior group”. Cluster one included students older than those in other clusters and who had their own specific dietary rituals. Meanwhile, cluster three included students younger than those in other clusters and who had high-level problematic mealtime behavior and a low preference for food. In particular, there were significant differences in age and food preference for each subdivided ASD group according to their eating behaviors.

As displayed above, the term “food selectivity” describes a wide range of behaviors or situations related to eating habits, such as restricted calorie intake, food refusal, food-related rituals or obsessions, behavioral problems related to mealtimes, preferences of certain foods, restricted variety of foods, and a diet restricted to specific categories of foods (dairy or protein-rich products). Also, the feeding problems in children with ASD can be associated with Pica disorder [24], atypical use of tools, preferences regarding food preparation [25], and the preference of foods according to texture, color, or temperature [26]. In fact, concerning the classification of children in relation to their mealtime behaviors, based on parent questionnaires, some researchers demonstrated the presence of different groups [27] as children with ASD were categorized as engaging in eating patterns of selective overeating, selective eating only, overeating only, or typical eating. Group differences were found in the areas of diet composition, BMI, and behavioral flexibility. Both the selective overeating group and selective eating only group were prone to favor calorie-dense, nutrient-deficient diets as compared to other groups. Eating groups also presented with differing profiles of everyday behavioral flexibility. These results suggest that selective overeating in ASD may present unique challenges and require tailored interventions.

### 2.1. Diet, Weight, and Nutritional Inadequacies

Firstly, the frequency of selective eating and nutritional deficiency was studied among 22 children with autism and an age-matched TDC group. Children with autism ate fewer foods on average than TDC. As compared to controls, children with autism had a higher average intake of magnesium, and a lower average intake of protein, calcium, vitamin B12, and vitamin D [28]. Another research team compared the nutrient intake from food consumed by children with and without ASD and examined nutrient deficiency and excess [29]. Successively, 3-day food records (N = 252) and BMI for children (2–11 years) with ASD were compared with both the National Survey data and a matched subset based on age, gender, family income, and race/ethnicity. Children with ASD and matched controls consumed similar amounts of nutrients from food. Only children with ASD aged 4 to 8 years consumed significantly less energy, vitamins A and C, and the mineral Zn; while those aged 9 to 11 years consumed less phosphorous. A greater percentage of children with ASD met recommendations for vitamins K and E. Few children in either group met the recommended intakes for fiber, choline, calcium, vitamin D, vitamin K, and potassium. Specific age groups consumed excessive amounts of sodium, folate, manganese, zinc, vitamin A (retinol), selenium, and copper. No differences were observed in the nutritional sufficiency of children given restricted diets. Children aged 2 to 5 years with ASD had more overweightness and obesity, while children 5 to 11 years were more likely to be underweight. More recently, an important study [30] involved a cross-sectional electronic medical record review to investigate the demographic characteristics, anthropometric parameters, risk of nutritional inadequacy, dietary variety, and problematic mealtime behaviors in a sample of children with ASD with severe food selectivity. Children (age 2 to 17 years) with ASD (N = 279), severe food selectivity, and complete nutritional data were enrolled. Successively, 70 children with ASD and severe food selectivity met the inclusion criteria and their caregivers reported 67% of the sample (n = 47) omitted vegetables and 27% omitted fruits (n = 19). Seventy-eight percent consumed a diet at risk of five or more inadequacies. Risk for specific inadequacies included vitamin D (97% of the sample), fiber (91%), vitamin E (83%), and calcium (71%). Children with five or more nutritional inadequacies (n = 55) were more likely to make negative statements during meals. Nevertheless, severe food selectivity was not associated with compromised growth or obesity. Likewise, a recent meta-analysis [31] examined the differences in nutritional intake and food consumption between children with ASD and controls, as well as the relative compliance with the dietary recommendations. The meta-analysis showed that children with ASD consumed less protein, calcium, phosphorus, selenium, vitamin D, thiamine, riboflavin, and vitamin B12, and more poly-unsaturated fat acid and vitamin E than the controls. The results also suggest a lower intake of calcium, vitamin D, and dairy and a higher intake of fruit, vegetables, protein, phosphorus, selenium, thiamine, riboflavin, and vitamin B12 than recommended. However, these results must be considered with care due to the low number of studies included in the analysis and the high heterogeneity. Additionally, another recent study evaluated the body composition, nutritional status through food selectivity and degree of inadequate intake, and mealtime behavior in children with ASD compared to neurotypical children [32]. A cross-sectional case-control study was carried out on 144 children (N = 55 with ASD; N = 91 with TDC) between 6 and 18 years of age. Body composition, nutritional intake, food consumption frequency (FFQ), and mealtime behavior were evaluated. As aforementioned, results showed a greater presence of children with a low weight (18.4% ASD vs. 3.20% TDC) and obesity (16.3% ASD vs. 8.6% TDC). The presence of obesity in ASD children compared to the comparison group was even higher when considering the fat component (47.5% ASD vs. 19.4% TDC). Moreover, children with ASD had greater intake inadequacy (50% ASD vs. 22% TDC), high food selectivity by FFQ (60.6% ASD vs. 37.9% TDC), and more eating problems (food rejection, limited variety, disruptive behavior), compared to neurotypical children. Concluding, children with severe food selectivity may be at an increased risk of nutritional inadequacies. As a result, we suggest monitoring nutritional inadequacies and implementing nutritional strategies to expand the variety of foods that children with ASD consume.

### 2.2. Hypotheses of the Multidimensional Phenomenon

Concerning the causes of food selectivity in children with ASD, scientists have examined diverse dimensions of altering feeding behaviors. As mentioned above, the importance of motor conditions and/or gastrointestinal complications should be addressed by clinicians. For example, scientists have found contrasting outcomes in gastrointestinal disorders (GID). A higher occurrence of GID may be linked with a more severe food selectivity in children with ASD [33]. Conversely, these two phenomena could not be associated [34]. Additionally, a recent research team investigated the prevalence of GID, food selectivity, and mealtime difficulties, and their associations with dietary interventions, food supplement use, and behavioral characteristics in a sample involving 247 participants with ASD and 267 controls aged 2–18 years [35]. Data were collected via a questionnaire. GIDs were observed in 88.9% of children and adolescents with ASD, more often in girls than in boys. High rates of food selectivity (69.1%) and mealtime problems (64.3%) were found. Food supplements were used by 66.7% of individuals, mainly vitamins/minerals, probiotics, and omega-3 fatty acids. In the ASD sample, 21.2% of subjects followed a diet, mostly based on gluten and milk restriction, including individuals exhibiting food selectivity. Frequency of GID, food selectivity, and mealtime problems correlated weakly, but significantly with behavioral characteristics in the ASD group, although not with food supplement use. Hence, this study demonstrated that a higher frequency of GID, food selectivity, and mealtime problems are common problems in preschoolers, school children, and adolescents with ASD, and together with dietary modification, they are significantly associated with ASD. To complete the description of the state-of-the-art of research, we report the results of a recent narrative review of the literature (the last 15 years) on food selectivity and its relationship with GID in children with ASD [36]. Mainly, sensory aversion in ASD leads to food elimination, based on consistencies, preferences, and other sensory issues. Consequently, the restriction of food groups that modulate the gut microbiota, such as fruits and vegetables, as well as the fibers of some cereals, triggers an intestinal dysbiosis with increased abundance in Enterobacteriaceae, Salmonella Escherichia/Shigella, and Clostridium XIVa, which, together with an aberrant immune response and a leaky gut, may trigger GID. It has been observed that food selectivity can be the product of previous GID. Likewise, GID could provide information to generate a hypothesis of the bidirectional relationship between food refusal and GID. On the other hand, immunologic dysfunctions have recently emerged as a factor associated with ASD. Although children with ASD are more likely to have GID, little is known about the association between food allergy and ASD. A cross-sectional study used data from the National Health Interview Survey collected between 1997 and 2016 [37]. Children aged 3 to 17 years were included while food allergy, respiratory allergy, and skin allergy were defined based on an affirmative response in the questionnaire by a parent. This research included 199,520 children (mean 10 years; 51% boys). Among them, 4.31% had food allergies, 12.15% had a respiratory allergy, and 9.91% had a skin allergy. A diagnosis of ASD was reported in 1868 children (0.95%). The weighted prevalence of reported food, respiratory, and skin allergies was higher in children with ASD (11.25%, 18.73%, and 16.81%, respectively) compared with children without ASD (4.25%, 12.08%, and 9.84%, respectively). In analyses adjusting for age, sex, race/ethnicity, family highest education level, family income level, and geographical region, the associations between allergic conditions and ASD remained significant. Finally, a significant and positive association of common allergic conditions—in particular food allergies—with ASD was found. An analogous study examined the parent-reported prevalence of co-occurring food allergies and ASD in a nationally representative sample of US children ages 2–17 in the National Health Interview Survey, study years 2011–2015 [38]. In the analytic sample of 53,365 children ages 2–17, there were 905 children with parent-reported ASD (prevalence of 1.7%) and 2977 children with parent-reported food allergies (prevalence of 5.6%). Parent-reported food allergies were nearly 2.5 times more common in children with ASD (prevalence of 13.1%) than in children without ASD (5.4%). These results indicate that food allergies commonly co-occur with ASD, which may have etiological implications.

Moreover, food selectivity could be due to problems in sensory processing (abnormal multimodal sensory responses). Concerning this main approach, food selectivity can be considered a manifestation of the altered sensory response and behavioral rigidity; yet food selectivity manifests itself through preferences regarding consistency (67%), appearance (58%), taste (45%), smell (36%), temperature (22%), [2]. Likewise, sensory processing anomalies, common in individuals with ASD, could be part of the possible mechanisms underlying food selectivity [15]. Essentially, children with ASD and food selectivity are hypersensitive to the consistency (soft, gelatinous, crunchy, hard); the taste; the smells (also of the people around them); the visual aspects (shape, color, and presentation); the temperature of the food (touching); and also to sensory stimuli that surround the environment in which the meal is consumed [39]. To date, sensorial anomalies are connected to autism diagnosis since such behaviors regard multiple sensorial stimuli and sense organs. The majority of autistic children could show hyper/hypo stimulation to tactile, gustative, olfactive, proprioceptive, and visual stimuli [40]. As a result, such sensorial dysregulation influences undoubtedly the mealtimes of children. As aforementioned, some characteristics of food (taste, smell, texture, temperature, colors) are shown by children in single or multiple combinations as well as in sameness, and some rigid behaviors can be displayed through the following stimuli as food presentation, cutlery, brand, and packaging [2,9,18,34,41,42,43,44]. Another study [45] compared oral sensory processing among children with (n = 53) and without ASD (n = 58), aged 3–11 years, examining the relationships between atypical oral sensory processing, food selectivity, and fruit/vegetable consumption in children with ASD. The results showed that more children with ASD had atypical sensory processing than children without ASD, highlighting how among children with ASD, those with atypical oral sensory sensitivity rejected more foods and ate fewer vegetables than those with typical oral sensory sensitivity. Recently, an experimental study was tempted to underline a connection between visual perception and food neophobia [46]. The present study examined whether children with ASD and TDC differed in their visual perception of food stimuli at both the sensorimotor and affective levels; a potential link between visual perception and food neo-phobia was also investigated. Subsequently, 11 children with ASD and 11 with TDC were tested. Visual pictures of food were used, and food neophobia was assessed by the parents. Results revealed that children with ASD experienced visually longer food stimuli than TDC. Complementary analyses revealed that whereas TDC explored more multiple-item dishes (vs. simple-item dishes), children with ASD explored all the dishes in a similar way. In addition, children with ASD gave more negative appreciation in general. Moreover, the hedonic rating was negatively correlated with food neophobia scores in children with ASD, but not in TD children.

Furthermore, other challenging behaviors have also been associated with food selectivity, such as aggression, choking, internalizing, and externalizing problems (anxiety or aggression), rejection of food, and repetitive and restricted behaviors. A study [47] of 256 children with ASD found a moderate but significant correlation between the Repetitive Behavior Scales-Revised [48] and the Brief Autism Mealtime Behavior Inventory (BAMBI), [49], indicating that children with more repetitive behaviors may also exhibit more problematic meal-related behaviors. Also, typical expressions of behavioral sameness could be: using the same utensils (cutlery or special dishes), paying attention to the presentation of the food (food contamination), accepting only certain brands, and paying attention to packaging. In synthesis, children with ASD and food selectivity insist more often than TDC with food selectivity to use the same dish (8% vs. 2%) or to request food prepared in the same way for each meal (28% vs. 20%) [50], and often have behavioral problems challenging to manage (for example, screaming, crying, irritability, aggression, escape, anxiety, turning the head, chewing without swallowing, spitting, and vomiting). A pioneering study observed that 70% of children with ASD in his sample showed food selectivity [43], currently this percentage varies across studies, although slightly lower than the above-mentioned.

### 2.3. Parental Stress in Autistic Children with Food Selectivity

In the research literature, parents of children with food selectivity report higher levels of stress than parents of children without food selectivity [34]. One study [51] evaluated the associations between food selectivity with behavioral problems during meals, marital stress, and influences on family members in a group of 53 children with ASD and 58 with TDC, all aged 3 to 11 years old. The results showed that compared to TDC, children with ASD were more likely to have high food selectivity, as a result of which their parents reported more behavioral problems during meals, with consequences that concerned greater marital stress and a strong conditioning on the eating habits of other family members. Furthermore, in response to these feeding behaviors, the caregivers may try to encourage/comfort their child, including reprimanding and eventually replacing dishes according to the preferences of the child. Consequently, the child learns to avoid undesirable foods by emitting challenging behaviors, as well as the parents involuntarily maintaining such behaviors by presenting only the preferred food at mealtime [52]. Also, the restricted food intake of the children can significantly influence the eating habits of the entire family, since children become irritable and exhibit some maladaptive behaviors such as tantrums, reluctance to sit at the table with family, or throwing and spit-out the food with interruptions to their typical meal routines [40]. Consequently, parents, to avoid challenging behaviors during mealtimes, tend to indulge their children’s food tastes by excluding certain foods from the eating habits of the entire family. For example, some families allow the child to eat separately from the family or provide the child with individualized support, supervision, and verbal redirection during meals [53]. However, when asked, the mothers claimed to be subjected to multiple sources of stress due to the impact of atypical eating behaviors and nutritional concerns on family dynamics [54]. Some mothers, interviewed about the perception of meals with their children with autism, described meal time as difficult and stressful, due to the child’s limited diet, emphasizing that behavioral intervention should be taken into account primarily to improve the eating habits of children with ASD, but also to reduce the perception of stress related to the meal routines of the families with children with autism and food selectivity [55]. The meal routine was experienced by mothers with a strong sense of responsibility, as they were often solely responsible for preparing food and meals and perceived this task as crucial to maintaining the child’s general well-being. Additionally, the need to negotiate a child’s feeding challenge was reported by many mothers as one of the reasons influencing family well-being and sometimes even the relationships with friends. It is therefore essential, for the clinician who is preparing a clinical intervention, to consider the need to also support the family through appropriate sessions of parent training. Finally, eating behaviors are also influenced by sociocultural dimensions [56,57], and educational styles implemented by caregivers can similarly have an impact on feeding behaviors. An explanation could be, a minor exposure of children to a varied range of foods provided, other than families could reinforce the dysfunctional behaviors of children during mealtimes [58]. Hence, environmental factors play a role in the development and maintenance of food refusal. The refusal of food can involve positive reinforcement in the form of attention (verbal reprimand or persuasion) or negative reinforcement, such as an early break from the meal, which also maintains the behavior of refusing food.

All the studies described so far seem to support the hypothesis of the multidimensional nature of food selectivity in children with ASD. Also, this phenomenon seems to be associated with important health issues in these children, as well as high levels of parental stress. For these reasons, food selectivity might reduce the quality of life of autistic children and their families, with possible detrimental effects on their future development. Therefore, food selectivity in children with ASD requires specific assessments and tailored interventions. In light of these considerations, in the following chapter, we describe a well-established direct/indirect assessment of food selectivity in children with ASD and related medical, sensory, and behavioral interventions.

## 3. Assessment

### 3.1. Nutrition Assessment and Feeding History

Concerning the nutritional assessment for children with ASD, clinicians should detect the causal nutritional inadequacies connected to the restricted intake, since children with ASD stereotypically choose diets such as junk snacks and candies limiting fruits and vegetables [19,49]. Consequently, nutrition assessment refers to dietary repertoire and corresponding nutrients consumed and recurrently excluded food categories influencing child development. Likewise, clinicians should gather anthropometric data, current diet, and related eating history. Evidence gathered throughout a comprehensive feeding diary includes food group intake, concerning fruit, vegetable, legumes, meat/dairy, and grains/cereals, and compares the child with well-known dietary recommendations. Specifically, clinical staff should ask parents for a weekly food diary investigating the current food categories accepted by the child as well as providing them with standardized questionnaires on feeding issues. On the other hand, since feeding problems could be influenced by gastrointestinal issues, a medical assessment to exclude this comorbidity is recommended so as to explore possible essential organic pathologies (food intolerances/allergies, diarrhea/constipation, reflux, and so on). Conversely, regarding low-functioning children, some infrequent behavior should be observed (without a clear antecedent) to identify gastrointestinal issues such as infrequent posturing, self-injurious and aggressive behaviors, bruxism, uncommon facial expressions, pica, and other unclear postures (also associated with dental problems). Similarly, since sensorial processing anomalies are shared between children with ASD, as well as over/under-responsivity and sensory seeking, an investigation of sensory stimuli having an impact on feeding behaviors (texture, temperature, taste, flavor, movement, and color) should be addressed.

### 3.2. Assessment of Feeding Issues

Firstly, before implementing an educational intervention, the following intersected aspects such as GID, assumed foods (food diary), and growth curves as suggested by Badalyan and Schwartz [59], should be addressed. Subsequently, a clinician could calculate the Body Mass Index (BMI) (kg/m^2^) of the target child monitoring it over time. Commonly, a dysfunctional diet henceforth could evoke eating problems such as obesity or being underweight; consequently, clinical staff should pay attention to these medical issues. Similarly, light has been shed regarding the impacts of the common GID, such as constipation (overall), vomiting, diarrhea, abdominal pain, and gastroesophageal reflux. Likewise, families could exclude some causal dimensions through medical examinations. On the other hand, families should investigate and share with clinical staff if the child assumes a gluten/casein-free diet and if he refuses some ingredients such as legumes, vegetables, carbohydrates, proteins, dairy, and so on. In our experience, therapists could consider reintroducing some foods that children consumed before they have shown challenging behaviors during mealtimes. To gather information concerning the current diet of the child, a well-recognized instrument such as FFQ (Food Frequency Questionnaire) could be considered [60]. This measurement permits the detection of the food categories that the child excludes, along with the related nutritional deficiencies. In fact, Bandini et al. [19] implemented the FFQ by comparing a clinical sample of children with ASD to controls. It is assumed that children with ASD should report more food refusal. Likewise, this evaluation permits the detection of a severe intake of single or specific nutrients, as a result of having the first hypotheses of the clinical intervention. The assessment indicates the first phase of a clinical evaluation process. It consists of a global assessment of the cognitive, emotional, and behavioral functioning aimed at recognizing the history and characteristics of the problem displayed by the individual and the impact on his daily life. Two different types of assessment can be distinguished: direct (through repeated direct observation in natural settings and/or interviews) and indirect (through the use of questionnaires, surveys, rating scales, interviews, and checklists). Finally, clinical assessment defines the target behavior and the variables that control dysfunctional eating behaviors.

#### 3.2.1. Standardized Questionnaires: Feeding Behaviors in Children

Starting with interviews of families, the clinicians could enrich their anamnesis of children using well-structured instruments such as questionnaires and measure scales. Conversely, planning the tailored educational program for children through this information comprises some biases. As a result, we should consider a combination of the assessment procedures, indirect and direct (also experimental if the case does not suggest a clear function of behavior). Hence, the questionnaire furnishes a perception of the caregiver on dysfunctional behavior during mealtimes (refusal, aggression, choosiness, rituals, selectivity, and so on), also indicating the occurrence/intensity of behavior and the relative stress and burden addressed by families. We have listed some well-known questionnaires in the following Table 1. Please, consider this list as a starting point to proceed with more detailed bibliographic research.

The Brief Autism Mealtime Behavior Inventory (BAMBI) is a tool composed of 18 items to identify the presence of aggressive and destructive behaviors during meals [49]. The first version of the tool consists of twenty questions, divided into three domains. After evaluating some weaknesses in the original version of the tool, the authors propose a version consisting of 18 items that evaluate the frequency of the child’s eating behavior on a 5-point Likert scale (1 = never; 5 = almost always). The items on the scale are grouped around three factors: “Limited Variety Factor”, which identifies the reluctance of children to try new foods or foods prepared in a different way concerning the texture and type of food proposed; the “Food Refusal Factor” dimension, aimed at evaluating the behaviors implemented by children to avoid unpleasant food intake; and the “Autism Characteristics Factor” dimension, which evaluates the most typical behaviors of the autism spectrum. Starting from the 18 items, Hendy, Seiverling, Lukens, and Williams [61] developed a new version of the scale, renamed the Brief Assessment of Mealtime Behavior for Children (BAMBIC), to evaluate the association between each dietary problem and characteristics such as age, gender, weight, and diagnosis in a group of 108 children with feeding problems, both ASD and TDC. BAMBIC revealed three subscales: “Limited Variety”, “Food Refusal”, and a third dimension that the authors named “Disruptive Behavior”. The tool also revealed the effects of the children’s gender, diagnosis, and age on eating problems. A subsequent study [62] validated the three subscales of BAMBIC on a sample of children without a previous diagnosis of behavioral problems. Summarizing, the first scale (Refusal) refers to the child’s refusal or denial of foods: “My child expels (spits out) food that he/she has eaten”. The second scale (Limited Variety) includes questions about a restricted consumption of specific categories of foods: “My child prefers the same foods at each meal”. The last scale (Autism Characteristics) highlights some challenging behaviors which can occur during the mealtime: “My child displays self-injurious behavior during mealtimes”.

The Behavior Pediatric Feeding Assessment Scale (BPFAS), the original instrument made up of 35 items, allows the calculation of four scores, relating to the four subscales of which it is composed [63]. Throughout the questionnaire, parents are asked to indicate, on a 5-step Likert-type scale, the frequency with which certain behaviors occur (from never to always), both in relation to the behavior of children and their own. The first 25 items concern the behaviors manifested by children during meals (Child Behavior Frequency), while the following 10 items concern the behaviors implemented by parents during the routine of meals with their children (Parents’ Behavior Frequency). There is also a second score applied to both sections of the test, a dichotomous scale (yes/no), which evaluates how much the child’s behavior is perceived as problematic (Child Behavior Problem) and how much one’s behavior is perceived as problematic (Parents Behavior Problem). Therefore, the BPFAS provides four scores are: Child Behavior Frequency; Parent Behavior Frequency; Child Behavior Problems; and Parent Behavior Problems. The Frequency Score scale reflects how often a behavior occurs while the Problem Score represents the number of eating behaviors considered problematic. High scores on both scales indicate dysfunctional food functioning. A Total Frequency Score (which includes the Parent Behavior Frequency Score and Child Behavior Frequency Score) greater than 84 is recognized as significantly higher than the average score and requires nutritional intervention. The results of the original study showed that, for children who indicated eating problems, the frequency and score of the problems were 2 SD above the mean, while no gender effect of the children on the Total Frequency Score was evident. The tool also proved to be particularly sensitive for capturing changes in the eating behaviors of children undergoing clinical intervention protocols aimed at reducing problems [64]. The BPFAS proved to be a tool variably correlated to observational data relating to the direct assessment of eating behavior, proving in fact to be the most reliable tool for assessing eating behaviors. Successively, Allen et al. [65] have applied a factor analysis to BPFAS by examining the original five-factor model. Therefore, a categorical exploratory factor analysis (CEFA) was applied to study the adequacy of the proposed model. Hence, the three-factor solution accounted for 43.13% of the cumulative variance. The authors respectively showed three factors: Food Acceptance, Medical/Oral Motor, and Mealtime Behavior. The three extracted factors were significantly intersected.

**Table 1 ijerph-20-05092-t001:** Descriptions of some main questionnaires regarding feeding behaviors.

ID	Description	Authors
BAMBI-C	Brief Autism Mealtime Behavior Inventory	[49]
STEPS	Screening Tool of Feeding Problems	[66]
CEBI-R	Children’s Eating Behavior Inventory-Revised	[67]
BPFAS	Behavioral Paediatrics Feeding Assessment Scale	[63]
PASSFP	Paediatric Assessment Scale for Severe Feeding Problems	[68]

#### 3.2.2. Sensory Profile

Since the sensorial anomalies have an impact on feeding behaviors, an assessment regarding the sensory profile in children is also suggested because the results may guide the intervention, avoiding time-cost failures. In the related literature, clinicians could find different well-studied instruments exploring sensory processing in children [69]. However, the Short Sensory Profile (SSP) is common among different studies since it is a thirty-eight-item standardized questionnaire of seven scales asking caregivers which aspect of sensory processing of the child is representative (tactile, taste, weak, seek sensation, smell, visual, movement), [70]. During a replication study, researchers showed that children with ASD reported differences in the majority of the items (more for tactile and auditory) than the controls [71]. Similarly, some authors discovered analogous results regarding preschool children with ASD and controls. Finally, and more recently, other authors showed a great impact regarding the seeking sensation, reflecting the before-mentioned sensorial modalities of hypo or hypersensitivity [72]. Since 2014, The Sensory Profile™ 2 has offered standardized tools (five questionnaires) to help evaluate a child’s sensory processing patterns in the setting of home, school, and community-based activities. The Sensory Profile 2 categorizes information by scores from sensory, behavioral, and school sections providing comprehensive reports of children. These questionnaires with the corresponding age range (ISP-2: Infant Sensory Profile 2, <6 months; TSP-2: Toddler Sensory Profile 2, for children between 7–35 months; CSP-2: Child Sensory Profile 2, for 3 to 14 years of age; SSP-2: Short Sensory Profile 2, for 3–14 years of age; SCSP-2: School Companion, for 3–14 years of age) offer caregivers and teachers evidence since sensory processing to help planning the intervention. For example, the 25-item ISP-2 gathers information from six sections (General, Auditory, Visual, Touch, Movement, and Oral Sensory Processing), while the other questionnaires include Quadrants such as Seeking, Avoiding, Sensitivity, and Registration other than Behavioural sections associated with sensory processing. Finally, the 44-item SCSP-2 includes four school factors: external supports, awareness and attention, tolerance, and availability (for a detailed overview, please see https://www.pearsonassessments.com/store/usassessments/en/Store/Professional-Assessments/Motor-Sensory/Sensory-Profile-2/, accessed on 5 March 2023).

### 3.3. Caregiver Feeding Practices

Since caregivers’ food preferences and correlated feeding practices could influence the eating behaviors of children, an assessment of these educational strategies should be considered before and during the intervention, including the parents in the implementation of the treatment, and ongoing parent training should be provided. Concerning some questionnaires that clinicians could adopt, we list two instruments: the Parental Feeding Style Questionnaire (PFSQ) [73] and the Caregiver’s Feeding Style Questionnaire (CFSQ) [74]. Firstly, the 27-item (PFSQ), measures (1) maternal emotional feeding, (2) instrumental feeding (using food as a reward), (3) prompting/encouragement to eat (more used by families with children with ASD), and (4) control over child eating [75]. On the other hand, the 19-item (CFSQ) identifies parent feeding styles in four categories: authoritative, authoritarian (high control), indulgent (child-centered), or uninvolved. Hence, these instruments help us to study the educational approach of the families orienting the intervention, including some functional useful strategies as behavioral interventions. For example, a parent could have a functional/dysfunctional physical and verbal approach towards the child to make them eat, as well as manipulate the food provided. The following Table 2 provides a list of typical behaviors of parents suggested also via the CFSQ.

### 3.4. Direct Assessment

Direct measurement regards the behavior exhibited by children during mealtimes. Although generally considered essential for treatment planning, this assessment is rarely used in group research since it takes a long time to implement. On the other hand, clinicians have utilized direct observation in a single-case design where functional analysis can be easily implemented [76]. For these reasons, studies using direct assessments of food intake behaviors in children with ASD during mealtimes are quite rare. In 1994, Munk and Repp [77] used a Behavioral Assessment procedure to identify the relationship between the characteristics of the food (type or texture) and the behavioral problems exhibited by five children during meals. Researchers manipulated only the antecedent conditions (characteristics of the food) and recorded the children’s responses to all conditions, identifying the following types of problematic eating behaviors: total refusal of food, acceptance of food about the type (type selectivity) with acceptance of some foods at the “full texture” and refusal of other types to the “full texture”, acceptance of food with a particular texture (pureed texture), and refusal to take the same food with a different texture (texture selectivity). Subsequent research [41] replicated the original study in a group of 30 children with autism and neurodevelopmental disorders, finding that 17 children had low acceptance, 9 had a moderate level of acceptance, and 4 had a high level of acceptance. Systematic Behavioural Assessment included 30-min sessions including four foods for each group (fruit, starch, vegetables, and proteins), providing the same foods for all children. Hence, eight sessions lasting five minutes were planned so that the examiner randomly showed each food of the four categories. Each session consists of 24 tests, within which the foods were presented at their natural consistency (table texture) and blended (puree texture). Finally, in a recent study, Aponte and Romanczyk [17] observed eating behaviors in a group of autistic children, by administering sixteen foods, evaluating those seven children who showed a high level of acceptance, the three who showed a medium level, and the eight low acceptances. Concluding, direct observation could reveal fundamental information about the stimuli that might potentially evoke sensorial or behavioral issues in children with ASD. This information might allow the implementation of more effective intervention programs to reduce food selectivity in children with ASD.

## 4. Clinical Interventions

The most documented multidisciplinary interventions provided by the health services address feeding issues at home, in day-hospital/centers or, for the most complex situations, in hospitalization.

### 4.1. Medical Interventions

Regarding medical issues, Pennesi and Klein [78] evaluated the efficacy of the gluten-free and casein-free (GFCF) diet in children with ASD. The authors found that the diet was effective in improving the behaviors of children with ASD showing gastrointestinal symptoms (in particular, constipation and diarrhea) compared to children without GID, suggesting that children predisposed to gastrointestinal abnormalities could benefit from a GFCF dietary intervention. In addition, Perrin et al. [79] explored the use of complementary and alternative medicine (CAM) in children with ASD. Parents of children were asked if their children received acupuncture, chelation, chiropractic, hyperbaric oxygen therapy, food supplements (vitamins, probiotics, antifungal agents, digestive enzymes, glutathione, sulfation, amino acids, or essential fatty acids), and special diets (classified as gluten-free or casein-free, and which do not involve the use of processed sugars). The study found that parents reported higher rates of CAM (in general and for special diets) when children also reported gastrointestinal problems. Finally, Mulloy et al. [80] conducted a systematic review of GFCF diets when implemented in the treatment of ASD, concluding that the published studies they identified do not support the use of these diets in the treatment of ASD. Instead, the authors identified some negative consequences for the use of the GFCF diet, identified in the reduction of cortical bone thickness and increased stigmatization, concluding that if a child with ASD shows behavioral changes apparently associated with the diet, professionals should consider assessing the child about the food allergies and intolerances in order to reduce the allergens identified in his environment.

### 4.2. Sensorial Processing Interventions

Excluding organic causes such as gastrointestinal problems, food selectivity can be considered a manifestation of the altered sensory response and behavioral rigidity, and an expression of altered brain networks [2,15,52,81]. The sensorial aspects play a crucial point in the development of problems related not only to the typical symptoms of autism, but also to secondary aspects, not least such alterations seen in the eating habits of children with ASD. According to the sensorial explanation, an evidence-based model as the Ayres Sensory Integration^®^ (ASI^®^, Saint Petersburg, FL, USA) [82] has been cited in the literature as “classical sensory integration” [83]. It is a type of intervention that aims to integrate sensory information of the children either regarding body perception or respect to the environmental stimuli through the use of specialized equipment and materials during targeted and playful activities, aiming to improve the adaptive behavior of children. Consequently, ASI^®^ is implemented by trained therapists, mostly occupational therapists (OT) in designed clinical settings. The therapy includes the active involvement of the child in naturalistic situations regarding specific arousal, attention, and motor programs. Individualized treatment occurs through weekly intensive treatment sessions to improve the target skills and to reach a greater skill level. Also, the therapy is based on fun interactions, offering a safe and highly sensory experience comprising the visual and auditory inputs, sensations on the skin, tastes, smells, and body balance to achieve different milestones in motor skills, adaptive skills, independence and self-care, cognitive abilities, executive functions (cognitive flexibility, planning, and working memory), communication, social skills, academic, reduction of problem behaviors (repetitive behaviors, stereotypies, and so on), fine and gross motor skills, and the regulation of the emotions. On the other hand, the Sequential Oral Sensory Approach (SOS) is a 12-week intervention program based on the typical developmental steps involved in nutrition [84]. This approach aims to “increase the range and volume of foods the child eats through play-based intervention” [85]. The approach, based on a systematic process of desensitization, is developed in six phases (visual tolerance, interaction, smell, touch, taste, and nutrition) and aims to guide the child through the exposition and experiences of a variety of foods and textures until he/she begins to interact, tasting a wider variety of foods [86]. Although it was not initially developed for individuals with autism, this approach is increasingly used to address the feeding difficulties experienced by children with ASD.

Also, a recent investigation compared a modified sequential oral sensory approach (M-SOS) to an ABA approach for the treatment of the food selectivity of six children with autism [87]. The authors randomly assigned three children to ABA and three children to M-SOS and compared the effects of treatment in a multiple-baseline design across novel, healthy target foods (multi-element design). Consumption of target foods increased for children who received ABA, but not for children who received M-SOS. Subsequently, the staff implemented ABA with the children for whom M-SOS was not effective and observed a potential treatment generalization effect during ABA when M-SOS preceded ABA. In addition, a study [88] examined the impact of a sensory intervention to address food selectivity in pupils with autism. The intervention was delivered in a school by teaching staff for a group of 19 children with ASD and difficulties in communication (aged between 4 and 10 years). The authors collected repeated measurements of the BAMBI, before and after the intervention. The results showed that the total BAMBI scores for the participants were significantly lower after the treatment than the baseline, also for both food selectivity and refusal, reducing the disruptive behaviors.

Regarding sensorial theories on clinical intervention, one study [89] tested a potential link between odor perception and food neophobia in 10 children between 6 and 13 years old and diagnosed with ASD, and 10 with TDC, by administering 16 stimuli. Food neophobia was evaluated by parents on a short scale. The results revealed that significant hedonic discrimination between attractive and aversive odors were observed in TDC and that the level of hedonic discrimination was negatively correlated with food neophobia scores in ASD children but not in TDC, highlighting a possible relationship between the hedonic reactivity of odors and eating behavior, a hypothesis that opens up new perspectives on the role of smell in the construction of eating behavior in children with ASD, corroborating the hypothesis that sensory processing, even of an olfactory nature, may play a crucial role in the alterations of eating habits. In line with the aforementioned study, another study assessed whether olfactory familiarization could render food odors more pleasant, and consequently food more attractive, to children with ASD [90]. Participants were first presented with a series of food odors (session one). Then, they were familiarized on four occasions (5 weeks) with one of the two most neutral odors (the other neutral odor was used as control) (session two). In session three, participants smelled the entire series of odors again. Both verbal and facial responses were compared from session one to session three. After session three, the children were presented with two identical foods (one containing the familiarized odor and one with the control odor) and were asked to choose between these foods. Results revealed a specific increase in positive emotions for the familiarized odor and that 68% of the children chose the food associated with the familiarized odor. These findings suggest that it is possible to modulate olfactory emotions and expand the dietary repertoire of children with ASD. A current study [91] evaluated food processing that alters certain sensorial aspects to solve food selectivity problems in a group of children with ASD. The study evaluated the effectiveness of the physical transformation of vegetables and fruits into snacks in order to improve the sensory perception of 56 children with ASD to the stimuli presented and allow a similar intake. The results showed increased consumption of fruits and vegetables for all three fruits measured and for 233 vegetables presented the idea that physical changes in foods can contribute to improving sensory processing and the consequent intake of foods otherwise rejected. The results of these studies are of considerable importance because they suggest that protocols for food selectivity cannot fail to take into account the aspects related to sensory processing and the processing of sensory information and that such information can be of fundamental importance in structuring targeted clinical intervention protocols and are therefore significantly more effective, but it also follows that behavioral interventions based on the principles of ABA are significantly more effective, especially when considering the aspects related to sensoriality, especially if visual and olfactory. Concluding, food selectivity in children with ASD can be improved by including strategies that address sensory processing.

### 4.3. Behavioural Interventions

The treatment of eating disorders in children is complex and necessarily multidisciplinary. A review of treatments for feeding disorders in TDC [92] described four main categories of interventions: behavioral, nutritional, oro-motor intervention, and parent training. Firstly, behavioral treatment aimed at reducing food selectivity in children often includes highly structured daily sessions, in which antecedent and consequence manipulation procedures are applied. The aim of these manipulations is both to increase the variety of the diet of people with ASD, and to decrease the frequency of problem behaviors exhibited at mealtimes or more generally exhibited by the child in relation to specific functions and/or situations. The most documented multidisciplinary interventions are provided in home settings with parents or more structured settings such as centers or hospitals. Similarly, a recent review [93] which analyzed day hospital or inpatient treatments for children with severe eating problems identified four points that characterize the most effective services in improved food intake: a multidisciplinary team of professionals; behavioral intervention to increase oral intake and at the same time manage behavioral difficulties related to meals; active participation and involvement of caregivers; and follow-up meetings. The objective of the behavioral intervention is to analyze the environmental antecedents and the contingent consequences of a specific problem behavior related to meals, considering sensory, motor, medical factors, and early traumatic events related to meals. Therefore, Applied Behavior Analysis (ABA) is the first-line intervention for the treatment of eating problems in children with ASD. In fact, data gathered during mealtimes suggest that ABA strategies for children with ASD and food selectivity are effective at school and at home regardless of whether these are implemented by therapists or caregivers [94]. These programs aim at reducing challenging behaviors by increasing the quantity and variety of food accepted by children and avoiding nutritional risks, through the following well-established behavioral practices: preference assessment, functional analysis, differential reinforcement, graduate exposure, escape extinction, stimulus fading, shaping, non-contingent reinforcement, simultaneous and sequential presentation, and mixing preferred and non-preferred foods [95]. The effectiveness of behavioral interventions was demonstrated in a survey [96] in which the authors randomly assigned six young children with ASD and food selectivity to an applied behavioral analytical intervention or waitlist check to assess the effects of a multicomponent, applied behavioral analytical intervention on the independent acceptance and mouth cleansing of sixteen new foods. The results showed that the percentage of independent acceptance and mouth cleansing increased for the applied behavioral analytical intervention group, but not for the waitlist control group until they implemented the intervention. In a recent study, the authors gave seven participants a choice between a change-resistant food and an alternative food during free- and asymmetrical-choice conditions [97]. Alternative-food consumption increased for two participants during asymmetrical choice when the feeder provided a preferred item for consuming the alternative food and no programmed consequence for consuming the change-resistant food. Alternative-food consumption increased for the other five participants after the feeder exposed at least one type of food to a single choice in which the feeder guided the participant to put the bite of alternative food in his or her mouth if he or she did not do so within 8 s of presentation. These results are important because participants consumed alternative foods even when their change-resistant foods were present, which is similar to typical mealtime contexts in which children have choices among foods.

#### 4.3.1. Stimulus Preference Assessment and Differential Reinforcement

The Stimulus Preference Assessment (SPA) is a relevant component of ABA programs because it allows the identification of the preferences of people with disabilities. The identification of the preferred stimuli allows therapists and caregivers to find potential reinforcers to be used in the implementation of educational interventions based on the use of positive reinforcement. Therefore, effective interventions can be implemented to increase functional behaviors to acquire new skills, and for the reduction of problem behaviors. In the literature, there are numerous studies of the different types of SPAs that can be grouped into trial-based assessments and free-operant assessments. In the trial-based preference assessments, the stimuli are presented in a series of trials. The results of these assessments are represented by the percentage of tests in which each stimulus is selected by the child, listing a ranking ordered by the preferences. Generally, a well-known applied SPA is the Paired-stimulus [98], where the stimuli are proposed in pairs so that each item is presented with all the other stimuli chosen for the assessment. An extension of the previous SPA is the Multiple-stimulus [99] in which six stimuli are presented, among which the participant chooses one for each test. This type of assessment was further extended in a study by DeLeon and Iwata [100] who developed the Multiple-Stimulus-Without-Replacement (MSWO), in which the stimulus selected in the set is removed at the next trial. On the other hand, Free Operant Assessments are based on answers that the children emit in an unstructured setting where the data can be collected every time; as a result, the play opportunities are not provided by others. Finally, the Brief Free-Operant (FO) [101] is a set of stimuli furnished for five minutes; the interaction of the child with the stimuli is recorded through a behavioral measurement through partial intervals of ten seconds.

After having identified the preferences and reinforcers of a child, a differential reinforcement can be implemented as a behavioral procedure used to increase the frequency of a specific behavior in a specified context, and decrease all other behaviors. For instance, this procedure provides contingent reinforcements to specific behaviors every time they occur and at the same time does not reinforce (extinguish) all the other responses. Following this procedure, the occurrence of the target behaviors increases. Consequently, the intervention identified as most effective is the Differential Reinforcement of Alternative Behaviors (DRA). Concerning feeding issues, differential reinforcement of food acceptance is implemented in association with Escape Extinction (EE) procedures as interventions addressing food avoidance behaviors [102].

#### 4.3.2. Escape Extinction

Several studies have shown that extinction is often a necessary component of effective treatments for food selectivity and rejection. Specifically, this procedure requires the child to take a specified number of bites before exiting the dining area and that the problematic behavior maintained by the escape does not lead to the end of the meal [103,104,105]. The EE procedure has been implemented in the form of non-removal of the spoon (NRS) by many clinicians until the child accepts a bite of the proposed food. In fact, the inappropriate behaviors shown by children are ignored, through extinction. Extinction in this specific case corresponds to the failure to deliver the escape (removal of unpleasant food) along with the acceptance of the unpreferred food, which are then reinforced with the presentation of preferred foods (differential reinforcement of incompatible behaviors; DRI) or objects previously identified as reinforcers. Subsequently, the variety of food proposed gradually increases, passing from one bite to more bites, or from a lower consistency in favor of a greater one, via demand fading which is associated with differential reinforcement which, correctly implemented, allows the systematic reduction of the preferred foods delivered to the child in proportion to the acceptance of non-preferred foods. Essentially, the magnitude of the reinforcement is reduced until the child chooses the favorite food at the end of the meal [106]. In summary, to decrease the frequency of behaviors maintained by avoidance, the EE procedure is implemented via the interruption of the escape contingency, maintaining the instruction until the child consumed the same amount of food, sometimes even with the pieces expelled [107]. There is also another variant of EE which consists of the session-interrupting criterion of eating the entire specified amount of food before leaving the session, and there is still another one that corresponds to the physical guidance of food acceptance for a refusal response [41,108,109,110]. However, although often necessary and effective, the EE procedure has been associated with high rates of inappropriate collateral behavior, especially in the early stages of treatment and when physical driving is a component; for example, crying or tantrums may arise; however, if non-spoon removal is successfully combined with antecedent manipulations or with reduced meal interruption criteria, the frequency and intensity of these adverse side effects are minimized [108,111]. It is important to point out that side effects caused by extinction procedures might include dangerous response topographies such as aggressive behaviors, response bursts, and response variability, which includes the emergence of other problem behaviors, and which could lead to safety problems for the client and those involved in the intervention. For this reason, the professionals who use these procedures should anticipate their possible undesirable side effects, and evaluate the therapeutic environment’s suitability, which includes the expertise and resources of those who manage the intervention [112,113]. Furthermore, extinction is rarely used as a sole intervention but is often combined with differential reinforcement procedures that are intended to augment alternative or other behaviors. 

#### 4.3.3. Stimulus Fading and Texture Fading

Stimulus fading is the process of the gradual removal of the prompt, suggestion, or support that the adult gives to help the child to emit an appropriate and independent behavior. In a well-recognized study [114], a reverse research design was used to evaluate the efficacy of the stimulus-fading procedure, along with differential reinforcement and extinction in order to increase the intake of calories via fluid by a 6-year-old child with feeding problems. The fading procedure was implemented to increase the concentration of a Carnation Instant Breakfast (CIB) along with milk and water. Since the researchers could manipulate the percentage of fluids blended, applying the principles of differential reinforcement and demand fading, they demonstrated that the stimulus-fading procedure was effective in increasing the quantity and type of food consumed via a gradual change in the characteristics of the water (bottle).

Another behavioral technique that should be considered by clinicians is the manipulation of food consistency [115]. Moreover, this behavioral technique could be associated with a reinforcement schedule contingent with an acceptance of the bit of food. Likewise, it could include the extinction of challenging behaviors (refusal/avoidance/escape). In fact, the aforementioned study showed an increase in the volume and consistency of food via texture fading. Mainly, clinicians should operationally describe the target behavior of the intervention (opening the mouth, touching the lips with the spoon). Subsequently, the dysfunctional behaviors should be defined (regurgitation, closing mouth, spitting out, chewing not swallowing, vomiting, and other problem behaviors). Furthermore, the authors indicate some data as the frequencies of the acceptances calculating the relative percentages. Finally, in addition to these behavioral data, the weights of the foods/bites before and after treatment should be measured. Concerning the same study, both the Stimulus Preference Assessment and Reinforcer Assessment were established. During the evaluation of the reinforcement, the first preferred stimuli identified via SPA were tested for the effectiveness of the reinforcement by providing access to the stimuli when the child was sitting on a chair. The stimuli which permitted the child to remain seated on the chair for a longer period of time were selected as reinforcements. Successively, each meal lasted 30 min or following an appropriate time in relation to the age of the child. Hence, the treatment consisted of praise or 15 s spent playing with some favorite toys, rewards that were contingent upon the acceptance of the proposed food or liquid. Moreover, an extinction of escape was also implemented (holding the spoon in front of the lips until the food was deposited in the mouth) for behaviors incompatible with the acceptance of the bite along with the extinction of the expulsion of the food (the spat food was newly put in the mouth of the child until it was swallowed). The meals began with the consumption of the amount of food recommended by a nutritionist at the initial texture; “probe meals” were conducted to determine the next texture for the fading. The textures presented were: 100% pureed, 100% junior, 100% ground, and 100% fine cut. Regarding the acquisition of the target behavior, target masterization per each probe was defined if the subject ate 80% of the food. If the probe meal did not meet this criterion, a second probe was furnished. In practice, the bites were subsequently presented as follows: 75% with the previous texture and 25% with the next one, also providing other alternatives as follows: 50% and 50%, 25%, 75%, and finally 100% with the next texture. Hence, the bites were presented using the chosen texture or texture combination until the criterion was mastered. If the lunch probe met the criterion for success, a lunch was conducted for the next texture. After reaching the criterion of 100% for the next texture, a sequence of probes was repeated. As a data collection, the post-treatment food weights plus the food weights of the expelled/vomited bites were then subtracted from the pre-treatment food weights to obtain the number of grams consumed. A multiple-baseline design across subjects was used to evaluate the effectiveness of the procedure. The results showed that a treatment consisting of texture fading, contingent reinforcement, swallowing, and the extinction of food rejection was effective in increasing the consumption of a greater textured food in a group of four children with food selectivity. However, the study found differences in the speed of fading among the children.

#### 4.3.4. Reinforcement: Positive and Negative

Currently, clinicians applying successful behavioral principles in clinical and educational settings for people with disabilities can refer to the science of respondent or operant conditioning. Therefore, positive and negative reinforcement influence increases the occurrence of a target behavior. As a result, these principles have also been implemented concerning feeding issues, especially in people with ASD. For example, negative reinforcement in the form of escape from eating has been hypothesized to be a primary function in maintaining eating problems. In this section, we describe some studies which have successfully applied reinforcement schedules for increasing feeding behaviors. Firstly, some authors have demonstrated that those positive reinforcements (contingent, positive stimuli such as edibles) could be more effective when they are compared with negative ones (escape) to foster the collaboration of children [116]. Generally, when the value of the reinforcement is well established by therapists, they increase easily the motivating operations of children. Likewise, a single case study [117] has investigated the individual and associated effects of the positive/negative reinforcement schedules regarding drinking through glass without EE. Therefore, a three-year-old boy eating three meals a day selected from jarred baby foods (refusing solid ones). Consequently, the main behavioral treatment aimed at increasing liquid assumption (one cup). Starting a behavioral treatment based on the reinforcement schedules, clinicians should plan a stimulus preference assessment to detect some possible reinforcement. Hence, in this study, researchers included a paired stimulus preference assessment of 15 foods, and a second preference assessment comparing most selected preferences such as peaches, carrots, and a 30-s break. Subsequently, the child had to choose one of three stimuli by cards. For example, If the child chose carrots, that food was presented on a spoon and was held there until he opened his mouth and accepted the bite. On the other hand, If the child chose the break card, he received a 30-s break from the cards and food. Therefore, the results of this second preference assessment helped to develop a treatment to increase cup drinking. In baseline, the therapist provided a cup and a verbal prompt (“Take a drink”), as a result, praise was delivered if the child consumed the drink. No differential consequences were provided for the expulsion of the food. During the treatment, a spoon of peaches was delivered following consumption of the drink during the positive reinforcement condition. Likewise, a spoon of carrots was delivered if the child showed inappropriate behaviors. Finally, the non-removal-of-the-spoon procedure was implemented. These results are unique because previously refused food items were used either as positive (peaches) or negative (carrots) reinforcement to increase cup drinking. In sum, the child increased acceptance of both peaches and carrots from 0% during baseline to 100%, the results of the preference assessments suggested that one previously refused food (peaches) became an appetitive stimulus and another food (carrots) remained an aversive stimulus.. Concluding, reinforcement schedules should be managed by therapists to increase their target feeding behaviors via SPA with children displaying challenging behaviors during mealtimes. Another study [118] showed that using individualized reinforcement and hierarchical exposure can increase flexibility in food intake in children with ASD. The results showed that after the intervention, all participants expanded their food repertoire and spontaneously requested new foods during follow-up, corroborating the effectiveness of behavioral strategies in reducing selective behaviors and increasing the quantity and quality of food taken by children with autism. A study [119] evaluated the effects of video modeling in the home on food selectivity in a group of three children with ASD, through the use of a home video modeling intervention during dinner for all three participants. The intervention consisted of two conditions: a video modeling condition and a video modeling condition plus a reinforcement contingency. The results showed that video modeling alone resulted in a greater acceptance of food than the baseline by participants, but when reinforcement was added to video modeling, higher levels of food acceptance occurred for all three participants.

#### 4.3.5. Simultaneous and Sequential Presentation

Commonly, parents with young children, especially those with special needs, experience difficulties in feeding their children. Other than the spontaneous delivery of possible reinforces, parents should try habitually to arrange foods through a combination/mix of opposite foods (accepted or not) or hiding refused ingredients into preferred foods. In scientific reports, the simultaneous and sequential presentation of preferred and non-preferred food to treat food selectivity has been addressed for several years [120]. Therefore, the mentioned clinical study confronted both these strategies (simultaneous and sequential presentation) concerning the consumption of the refused foods for three children. During the simultaneous clinical condition, the preferred foods were furnished in combination with no preferred foods (for instance, broccoli on a potato chip). On the other hand, the sequential methodology comprised the temporal contingency between the food acceptances of the previously refused food and an immediate provision of the preferred food. Consequently, the simultaneous presentation showed greater progress for two participants while one child showed similar results, requiring additional physical guidance and a representation of the bit of food. Finally, the sequential presentation displayed significant outcomes for only one child after many trials. Successively, the simultaneous procedure was performed by placing a favorite food above or below the non-favorite food; or by immersing the non-favorite food (e.g., strawberry) inside the favorite food (chocolate), until it was completely covered, while the sequential presentation consisted in giving the child the favorite food only after he has accepted a non-favorite food [121]. Concluding, the research regarding both educational feeding strategies for children with food selectivity should be considered, along with the other effective behavioral techniques that are associated.

#### 4.3.6. Physical Guide

Commonly, physical aids are regularly implemented during a rehabilitation intervention with autistic children and cognitive impairments to reach some developmental skill either in social behaviors or regarding autonomies. For this reason, different physical prompts can be provided through well-established strategies such as most-to-least prompting along with reinforcement schedules and the fading of the prompts. Likewise, these behavioral methodologies as prompt fading are fundamental in evoking a target behavior without the assistance of the tutor. Concerning feeding issues, additional treatment components were evaluated in the above-mentioned study [120]. Since the simultaneous condition did not display evidence for a child, the authors introduced a physical prompt during the trials. Hence, the behavioral therapist placed food in the child’s mouth without reintroducing the bites spat out. However, the food refusal did not decrease, and as a result, the representation of the expelled bites was required (both preferred and non-preferred food were represented when they were spat out). If the child expelled the non-preferred bites, only the non-favorite food was reintroduced. On the other hand, during the sequential presentation, the favorite bites were furnished when the child accepted the previously refused food. Finally, the authors demonstrated that simultaneous conditions obtained more evidence than sequential ones for all the children. Suggesting a possible explanation of these results, when a child tastes aversive foods along with a preferred one, the flavor could become acceptable than a single presentation. Nevertheless, the physical guide (commonly associated with EE) could be necessary when the only differential reinforcement does not show a significant outcome. However, it is very important to emphasize that those who perform these procedures should be well trained and should know how to minimize the risk of injury; they should, in fact, always consider the risk of suffocation, not putting a spoon in the open mouth of a child, for example, while they are crying; likewise, they should bear in mind the difficulty in using oral-motor skills effectively when the head is tilted back. One must always be very careful in using physical guidance as, although it is typically seen as an escape from the extinction procedure, it could also be seen as a punitive procedure. The scientific literature demonstrates that this procedure is used when a child exhibits near-zero levels of food acceptance after the demands of the feeding task have been significantly reduced. Therefore, for these reasons, physical guidance is used as a last resource and is generally used in conjunction with antecedent manipulation procedures [108].

#### 4.3.7. Shaping

Shaping is defined as the differential reinforcement of a target behavior. When applied to the feeding behaviors, during the shaping procedure the tutor should gradually reinforce the successive approximations of the target feeding behavior, such as: placing food in the spoon, food near the lips, food near the tongue, chewing the food, and so on. A study evaluated the use of shaping in increasing food consumption in two children with autism and ADHD [122]. Furthermore, the measures of the dependent variable were the level of food acceptance (total refusal, touching food with the lips, bringing food to the mouth, and chewing food) and the number of new foods assumed (variety). Firstly, a Paired Choice Preference Assessment was conducted to identify the two preferred stimuli. Successively, a combined multiple-baselines procedure was implemented. In the baseline, the foods selected in collaboration with the caregiver were presented. The participant was told, “eat your snack”, without providing any other instructions or any intervention on the consequence neither in relation to the consumption of food nor in the presence of escape behaviors. In the intervention phase, the trials were identical to the baseline sessions, only the targeted level of food acceptance was followed by the reinforcement. Moreover, some colored signals were used to communicate to the child that those foods were associated with the reinforcement (for example, foods in colored or white muffin cups). As a result, food placed in a colored muffin tin was associated with differential reinforcement by the therapists. Hence, if the participant responded correctly following the acceptance level within 2 min (or a higher amount of time), they would have received the 30 s of access to the preferred stimulus. In addition, the criterion for obtaining reinforcement was raised to the next level in the shaping hierarchy as the participant completed the mastered criterion until he reached complete acceptance. Subsequently, a final phase was implemented in which more foods had to be consumed. For example, the consumption of more foods was gradually reinforced, and the duration of the trial depended on the number of foods that the child had to consume (for instance, four minutes to consume two foods). The results showed that in the baseline phase, the refusal level of food was higher than 92% for both children; conversely, with the introduction of the intervention, improvements in food acceptance levels were rapidly demonstrated, highlighting constant improvements for all the food provided. Additionally, in a recent study, researchers examined the effects of a response-shaping procedure using a large rotating food set and a small constant food set on food acceptance for two boys with ASD [123]. The small set consisted of three foods that were presented during every session; the large set consisted of fifteen foods, of which three were presented during each session, in randomly ordered sets. Researchers measured the percentage of correct behaviors and the cumulative number of foods with which participants interacted. Two concurrently operating, multiple-baselines-across-behaviors designs were used to assess whether the shaping procedure resulted in increased correct responding compared to the baseline conditions and whether the intervention was differentially effective with large versus small food sets. The procedures were similar in efficiency for one participant, although he ate many more foods in the large set condition. For the other participant, shaping was successful at increasing some acceptance behaviors (e.g., putting food in his mouth), but only the small set resulted in eating new food. Hence, practitioners should consider the use of less restrictive or intrusive interventions to promote food acceptance and the use of larger sets of foods, modified to include fewer foods in the case of a poor response to intervention.

#### 4.3.8. Non-Contingent Reinforcement

Although EE is defined as the necessary component for treatments aimed at increasing food acceptance and reducing problematic meal-related behaviors, clinicians should consider other behavioral strategies. Starting with some considerations, the differential reinforcement, when implemented alone, could be ineffective in increasing food acceptance; in these cases, escaping extinction should be implemented to increase food acceptance. On the other hand, Reed et al. [124] found similar results with the application of non-contingent reinforcement (NCR). Since extinction often produces side effects such as aggression and emotional behavior, differential reinforcement and NCR can be more useful in reducing problem behavior for some participants. In fact, a subsequent study, Allison et al. [125] has compared differential reinforcement with EE and NCR with EE directly to determine which of both combinations was most effective in increasing food acceptance and in reducing challenging behaviors in a child with autism. A combined multi-element and ABAB inversion design was applied to evaluate the effects of the differential reinforcement of alternative behavior (DRA) with EE versus NCR and EE in relation to food acceptance, problem behaviors, and negative vocalizations. During the baseline of the treatment assessment, the food remained in front of the child’s mouth for 30 s, while no consequence was provided if the child did not accept or expel the food. During DRA with EE, the bite remained on the child’s lips until the therapist was able to deposit it in his mouth so that the therapist presented a 30-s access to the favorite toy and praise. The next bite was presented after the 30 s reinforcement interval if the child had a clean mouth. The therapist did not provide differential consequences for the problem behavior but blocked it if necessary. On the other hand, the expelled bites were represented until the child swallowed for a maximum of 20 min. Conversely, the NCR with EE combination was similar to the DRA with EE condition except that the same preferred objects and therapist attention were available throughout the entire session. The results showed that both treatments increased food acceptance to 100%, reducing problem behaviors related to the escape. Concluding, the two procedures are equally effective in increasing bite acceptance, reducing problem behavior, and minimizing negative vocalizations. According to the authors, the caregiver’s preference for the two interventions should be considered, also for the maintenance of the treatment. An important aspect was that the social validity questionnaire revealed that the mother preferred NCR as more acceptable, easier to implement, and more suited to the child’s needs because the mother felt that continued access to favorite toys was more “comforting” for the child than providing toys after a food acceptance.

### 4.4. Parent-Mediated Intervention and Parent Training

The clinical interventions for children, adolescents, and young adults with autism have demonstrated gains in different developmental domains where parent inclusion and related training have been largely suggested and recommended as evidence-based treatments, obtaining progress in different developmental skills of children with ASD, especially for social behavior and communication [126]. Although there is a consistent literature on parent inclusion in comprehensive and focused approaches, there is a smaller amount of information about the efficacy of the Parent Implemented intervention (PII) in which parents provide feeding interventions in their home. A comprehensive synthesis of the outcomes in feeding interventions indicated that only 58.3% of the interventions documented a provision of parent training and that over 80% of the studies had trained professionals rather than parents. Nevertheless, there were studies that demonstrated the effectiveness of behavioral interventions when implemented by parents [127]. In Anderson and McMillan [128], a parental use of EE and differential reinforcement to treat food selectivity has been addressed. A child, a five-year-old boy with a pervasive developmental disorder and severe intellectual disability, consumed only mashed potatoes, yogurt, and pureed apple. Organic or physiological causes were excluded. All sessions were led by the parents, who fed him with a spoon. The fruits were targeted for the intervention as requested by the parents. An assessment of food preferences was not conducted; however, differences in acceptance between favorite foods and fruit suggested that parents were able to accurately identify their favorite foods. Therefore, parents followed training to use DRA (preferred foods) along with EE. Moreover, parent training comprised verbal and written content and video tutorials. Finally, feedback was delivered and gradually delayed during weekly supervision. Additionally, the parents received supervision on video-recorded trials regarding adherence to the intervention. During the first application of DRA along with extinction, the child showed several improvements in food acceptance, even if the parents showed scarce adherence concerning the integrity of the clinical protocol. In fact, the parents showed more ability in reinforcement schedules than EE. Undoubtedly, parents that address challenging behaviors of their children during mealtimes, such as in other circumstances, could show apprehension and doubts regarding an authoritative educational style to follow; as a result, problem behaviors such as food refusal could be accidentally reinforced.

Starting with these considerations, tailored parent training on feeding issues enables caregivers’ acquisition of various competencies by receiving correct clinical supervision. Lastly, the video feedback used for the consultation could be implemented as an effective supervision technique, paying attention to the provision of supervision in time, since it should reduce the low adherence of the parents to the procedure. Finally, wherever possible, a clinician could implement immediate supervision or a mixed model (in person, at a distance, and/or via video feedback); additionally, the organization of the treatments should consider more ethical issues as correct as well as frequent supervision, since the children who do not receive a steady consultation about their treatment could waste days, weeks, and sometimes months before reaching the acquisition of the target behaviors, also concerning feeding issues.

Generally, parental involvement can change depending on familiar resources and the type of the intervention (e.g., behavior-based or sensory-based approaches) and providers (e.g., occupational therapists, psychologists, or speech therapists). On the other hand, a study [105] evaluated the effects of Behavioral Skills Training (BST) on the parental treatment of children with ASD, where parents implemented a food selectivity treatment package that consisted of repeated taste exposure, escape extinction, and fading, with a multiple-baseline design across mother–child dyads. The study involved three dyads of mothers–children with ASD. Parents had attempted unsuccessfully to implement a home-based plan involving access to preferred foods following the acceptance of non-preferred foods. The study recorded the percentage of correct steps performed by mothers who were instructed to present the bites and increase their amount. For child behaviors, the acceptance of pea-sized and half-spoonful bites of foods within the 30 s and the number of bites of foods with disruptive responses during taste sessions were recorded. The procedure involved a preliminary assessment in which the experimenter asked the mother to indicate on a list the foods consumed by the family and not by the child. In the parent training phase, firstly the experimenter read the task analysis of the taste session, then he modeled two taste sessions with the child and asked the mother to try the same procedure, providing three comments related to the execution corrected and two comments related to incorrect execution, repeating this cycle of modeling, testing, and feedback when necessary. The mother then performed three tasting sessions without feedback from the experimenter, who evaluated the correct executions when the child ate the bite. When the mother successfully averaged at least 90% of the steps, the experimenter began the probe meal training, with the same procedure. After the mother had received feedback, she performed a second evaluation of three tasting sessions followed by a probe meal, following which, if she correctly averaged at least 90% of the steps, the training was considered complete. The results of the study showed that the mothers’ baseline performance for the tasting sessions averaged less than 50%, while their performance for probe meals during the baseline period averaged at least 70%, while the percentages of correct steps performed were significantly higher after training, highlighting improvements in each mother’s performance for both the tasting sessions and probe meals during post-training and follow-up. During the basic tasting sessions, all children consistently refused bites and engaged in feeding interruptions, while during post-training, each child showed an increase in accepted bites within the 30 s and a decrease in feeding interruption. Also, results were maintained during post-training and follow-up probe meals, confirming an increase in the number of foods consumed by their children after treatment and evaluating the effectiveness of the BST package as excellent, considering the modeling component of BST to be very useful. A subsequent study [129], evaluated a video modeling strategy together with feedback provided in vivo to teach three parents of children with autism to implement a strategy aimed at reducing their children’s selective eating behavior. The results showed that for one parent, video modeling proved to be a sufficient procedure, while the other two parents needed in vivo suggestions and feedback. In more recent years, the literature internationally has provided some approaches for including families in feeding clinical interventions. Firstly, a study evaluated the effectiveness of a family-centered feeding intervention, Easing Anxiety Together with Understanding and Perseverance (EAT-UP™), for promoting food acceptance of children with ASD at home [130]. A concurrent, multiple-baseline design was used with systematic replication across three families. The baseline was followed by an ‘Intervention-Coaching’ phase and then an ‘Intervention-Independent’ phase. Using direct observation and pre- and post-intervention questionnaires, data on acceptance of less preferred foods and challenging mealtime behaviors were collected. Procedural fidelity was monitored throughout all study phases. Data were analyzed using visual analysis and measures of effect size. All children demonstrated increases in food acceptance and dietary diversity and decreased challenging behaviors. Likewise, a recent therapy model has combined two treatments—the Family Based Treatment (FBT) and the Unified Protocols for Transdiagnostic Treatment of Emotional Disorders in Children and Adolescents (UP-C/A)—for young patients with ARFID plus ASD, which allows clinicians to personalize care based on each patient’s unique presenting need [131]. This review presents two distinct cases which showcase the use of the FBT + UP for ARFID approach for treating comorbid ARFID and ASD in a clinical setting. Case one demonstrates the application and reliance on FBT, while Case two draws upon UP to facilitate behavioral change in the patient. Finally, the Managing Eating Aversions and Limited Variety (MEAL) Plan is a structured parent-mediated intervention for children with ASD and moderate food selectivity [132]. A reported group-based clinical trial revealed a positive treatment response rate of 47.3%. Although encouraging, this response rate raises questions about factors that may affect treatment outcomes. This research examines the impact of child and parent characteristics and feeding behaviors on treatment response. Higher maternal education and higher child communication abilities at baseline were associated with positive treatment responses. Improvement in sitting at the table and reductions in disruptive mealtime behavior promoted treatment success. Results also suggest that an individually delivered MEAL Plan may offer more flexibility than group-based interventions for some parents.

In the current section, we have indicated various recommendations to implement parent training for families having an autistic child with food selectivity. Firstly, clinicians should gather screening and detailed information concerning the child and ongoing interventions, as well as information about food preferences and educational parenting strategies. Nevertheless, clinical staff should assure that families have enough resources to begin a clinical intervention with their child. In these cases, staff could consider including the family during the intervention or generalization. The format could include group or individual settings to address similar feeding problems, as well as internet-based, home/community, or mixed training methodologies. Nonetheless, we would please suggest some research-based parent training strategies. Mainly, a well-established practice such as BST involves either individual or group settings providing instruction, modeling, role-playing, demonstration/trials, and feedback. Also, parental skill acquisition should be collected via an integrity data sheet showing the reported progress of the parents. Generally, some behavioral strategies can be included, such as delivering contingent praise to correct demonstrations of parental skills and correction for incorrect ones. Moreover, during a discussion, a trainer regularly presents information to an audience (single family or group), guiding the conversation to show the clinical program. Similarly, immediate coaching comprises verbal and/or nonverbal information during or immediately after a parent skill exhibition. Coaching and performance feedback can be delivered in different methods, including in person or through teleconsultation. For example, during an in vivo modeling, a trainer demonstrates some functional educational practices commonly in the family’s home, where the parent has to immediately imitate the model shown. Furthermore, parent training courses generally include didactic information where the trainer furnishes a handbook, checklist, exercises, and group play activities to increase the knowledge of the participants. Likewise, the modules of the course could be connected with individual needs. As mentioned above, clinicians should also consider the inclusion of innovative technologies to deliver motivational instructions such as computer/online activities, virtual reality settings, and interactive/educational games, as well as telehealth consultation [133]. Correspondingly, role-playing is a behavioral rehearsal involving a parent showing a learned procedure along with another participant. Likewise, the role-playing activities for practitioners (e.g., therapists, assistants, and psychologists) should be considered more ethics-based than in-vivo training to furnish a coherent and stable program to the children,. Finally, video modeling is a well-known evidence-based practice showing a video demonstration of a skill intended to teach or cue a parent to perform/imitate the skill. Then, video modeling can include either another person showing the expected skill or the same parent performing the ability in a video. Concluding, the content of the parent training could be comprehensive (including several developmental skills of children) or skill-focused (food refusal, challenging behaviors, autonomies, social communication, and other developmental skills) in the case of different clinical targets. Clinicians should provide correct information about the current evidence surrounding the intervention, giving free choice to parents, while reporting the following point-to-point resources as described in this narrative review: theories about food selectivity, clinical assessment, parent training, health impairments, sensorial, medical, and behavioral interventions.

## 5. Discussion

The current narrative review sheds light on how to assess food selectivity in children with ASD and how to plan effective and individualized interventions. Food selectivity has been defined in the scientific literature as food refusal, restricted variety of food intake, and in severe cases single-food (or liquid) intake [19]. Autistic children display a greater prevalence of food selectivity compared to TDC. Children with autism tend to refuse more vegetables, legumes, and fruit than other foods. In very extreme cases, some of them tend to accept only fluids or pureed textures. Commonly, choosy eaters frequently refuse certain types of foods, preferring specific food categories such as junk snacks (resulting sometimes in a reduction or significant weight gain). Undoubtedly, feeding problems become a severe burden for families and therapists assisting children with ASD with comorbidity of feeding issues. Since a restricted diet can influence the nutritional intake of children, causing significant stress to the families [32,51], feeding problems should be considered as a severe burden for families and therapists assisting children with ASD with comorbidity of feeding issues.

During the last decades, several authors investigated food selectivity in children with ASD. Also, numerous studies investigated the effectiveness of various intervention programs. The analysis of these studies indicates that, currently, clinicians should assess the possible causes of feeding issues before the implementation of an intervention targeting food refusal behavior in autistic children, such as GID, diet, allergies, motor and dental problems, sensorial dysregulation, challenging behaviors, and finally the feeding educational styles of parents [2,33,35,134]. This is particularly of interest, since many authors demonstrated that children with ASD can display more sensory abnormalities than neurotypical children, which further influences their feeding behaviors (taste, texture, color, movement, and temperature). Undoubtedly, a sensorial reticence to try new food or no preferred food hardly influences the food acceptance of the children. In fact, during mealtime, there might be several sensorial issues causing hypo or hyper reactions as we mentioned before in this review [2,9,18,34]. Thus, the association between sensorial processes overall for autistic children and food refusal has been well-established. Moreover, the scientific literature on the association between sensory desensitization and the increment of frequency/volume of foods is offering some useful clinical protocols to address feeding issues [84,87]. However, these studies are still rare, despite their importance. For these reasons, more studies extending these research lines are requested. Even if the behavioral program seems the first-line approach to reduce the feeding-dysfunctional behaviors at mealtime for children with ASD, interventions should consider including sensorial aspects, especially when the origin of the refusal is essentially sensorial. For example, in the study of Peterson et al. [96], a subsequent intervention that was ABA-based increased the generalization of the previous M-SOS program. Consequently, clinicians should pay attention to tailoring the clinical protocol based on the antecedents of the feeding issues for every child (medical, familiar, sensorial, and behavioral dimensions) to boost the efficacy of the intervention.

As mentioned, food problems in children with ASD could also simply occur due to challenging behaviors (splitting out the food, escape, the occurrence of sameness regarding the presentation of the food, self-injury or aggressive behavior, scarce interests, and so on). Especially, the behavioral literature regarding dysfunctional behaviors during mealtime demonstrated that the function of food refusal can be identified in the escape or by social attention [14]. The literature on food selectivity in children with ASD has demonstrated that those behavioral interventions significantly increase the food intake of children with ASD [135], showing the most effective services in improved food intake, such as: a multidisciplinary team of professionals, behavioral intervention to increase oral intake and at the same time manage behavioral difficulties related to meals; active participation and involvement of caregivers; and follow-up meetings.

In the section concerning behavioral interventions, we have listed diverse, effective techniques based on ABA principles displaying some examples of their implementation (the clinician could replicate or integrate a published clinical protocol). Mainly, a practitioner should receive complete training to implement an ABA therapy for feeding issues, and a task list regarding behavioral skills requirements is warranted by the Behavior Analyst Certification Board (BACB; https://www.bacb.com/, accessed on 9 January 2023). Additionally, these treatments need intensive supervision from a Board-Certified Behavior Analyst (BCBA), which assesses the child with the family and clinical staff to plan a suitable intervention program for the child. Generally, some BCBAs are more expert than others regarding feeding issues, as well as offering training to clinical staff; consequently, families need to collect this information when they ask for support for their child. Likewise, the family needs to receive training to be included in the intervention. As mentioned in various sections, behavioral techniques have demonstrated an advantage in incrementing food intake in children with ASD displaying severe food selectivity (differential reinforcement, shaping, stimulus/texture fading, simultaneous/sequential presentation, positive/negative/non-contingent reinforcement, along with escape extinction). Conversely, some behavioral techniques should be taken into consideration more than others, since someone can generate an increase in challenging behaviors such as physical guidance/pressure, punishment, and sometimes extinction. Another limitation of a behavioral, staff-mediated intervention is the generalization of the targets acquired; necessarily, the parents should maintain the level reached by the children with the staff. Since education and modeling received by the caregivers could also have an impact on children’s feeding routines, either through parents’ food preferences or through parental feeding practices, parent training and parent inclusion in the feeding therapy are recommended. In the section on parent training, we have suggested monitoring the progress of parents (fidelity) in implementing the behavioral program.

Regarding some methodological limitations to the current state-of-the-art in this research area, we report some recurrent issues regarding the research concerning food selectivity. Firstly, many authors used different definitions of food selectivity over time [11,12,19,43]. As a result, this incongruence could have caused some differences in data collection. Likewise, diverse direct or indirect measurements and a major focus on various dimensions (nutritional, medical, behavioral, educational, and so on) could have slightly influenced the research results. Hence, an advantage of the current narrative review is to describe the research in every aspect of the phenomenon. Similarly, some researchers have generally classified choosy eaters with ASD in three or four clusters via the interaction of some dimensions [23,27,56]. For example, the first authors claimed only one out of three clusters was described as high risk for feeding issues, which included low levels of BMI and high levels of GID, taste/smell/visual hypersensitivity, and dysfunctional behaviors at mealtimes. Successively, Park et al. [23] displayed the analysis of mealtime behavior resulted in a classification into three clusters: from “low-level” to “high-level” problematic mealtime behavior (the last cluster included younger students, high-level problematic mealtime behavior, and a low preference for food). Finally, children with ASD have been categorized as engaging in eating patterns of selective overeating (high risk), selective eating only, overeating only, or typical eating. Group differences were found in the areas of diet composition, BMI, and behavioral flexibility [27]. Comparing these theoretical classifications, it is possible to detect some similar insights. Firstly, all the clusters were different for some of the well-established variables explaining the phenomenon (diet, allergy, nutritional intake, GID, BMI, sensory process, dysfunctional behaviors, and parental feeding practices). Concluding, all the authors argue that only one cluster could be defined as high-risk. To sum up, even if in a clinical sample only a subgroup can be considered “severe” regarding feeding issues, the research has shown that if a child does not receive early intervention, he or she tends to display the problem over time. On the other hand, these clusters examined by authors permit clinicians to immediately detect the core symptom of a child along with the correlated variables (nutritional, medical, sensorial, behavioral, and environmental). However, the classification in subgroups could permit clinical staff to assess the efficacy of treatment with respect to the children’s profile in the baseline (tailored treatment). For example, if a child is characterized by more sensory issues than educational ones, a sensorial propaedeutic intervention could be suggested before behavioral intervention. Likewise, in the case of a child displaying a severe nutritional intake, an interdisciplinary staff including the physician, nutritionist, and psychologist should address both medical and behavioral comorbidities to immediately reduce the nutritional insufficiencies of the child.

In addition, we described both direct and indirect instruments (as interviews and standardized questionnaires) that clinicians can include to obtain a detailed assessment of the processes involved in food refusal in children with autism. This comprehensive assessment might allow clinicians to detect not only the influence of individual and contextual factors, but also eventual comorbidities. Also, a direct assessment permits the effective detection of the function of the feeding behavior [17]. This information could help clinicians to identify the target behaviors of the intervention, along with the contextual factors that might constitute specific advantages and possible barriers to adequate food intake. Likewise, clinicians should include an assessment of parental feeding practices, including a profile of the family stress and engagement.

Another important direction should consider the possible role of gut microbiota in food selectivity [136]. GDI symptoms in ASD have been extensively studied during the last decades, showing a higher frequency of food selectivity, mealtime problems, and GID in individuals with ASD, for a review, see [36]. These results give rise to the hypothesis of a bidirectional relationship between food refusal and GID in this population. Namely, food selectivity and mealtime problems might reduce the assumption of the kind of foods that modulate the gut microbiota, such as fruit, vegetables, and fibers. For this reason, children with ASD might be at a higher risk to develop intestinal dysbiosis and related GID compared to peers with a more balanced diet [137]. At the same time, children with ASD might present, very early on in development, an intestinal dysbiosis, which is known to affect brain development and behavior through the neuroendocrine, neuroimmune, and autonomic nervous systems [138]. Thus, food selectivity and mealtime behavioral problems might emerge late in development as correlates of sensory aversion and other characteristics related to the ASD condition. Certainly, more studies are needed to better clarify the nature of this relationship between gut microbiota and food intake behavior in individuals with ASD. It might be of interest to evaluate intestinal dysbiosis in newborns as a possible biological marker of ASD, which might be detected well before the food selectivity and mealtime behavioral problems that will emerge later in development. Also, early intervention to prevent intestinal dysbiosis by introducing appropriate microorganisms in a well-balanced diet should be considered very early on in development. Future research needs to address some crucial methodological issues which led to controversial results in previous studies, such as a high heterogeneity in the definitions of GDI, in the selection of the sample, and in the diagnostic tools, which also have been mainly quite indirect measures, such as parent reports [139].

According to the studies, we proposed three clinical approaches: medical, sensorial and behavioral. The first described approach includes studies about diets, allergies, and supplements as well as gastrointestinal issues. These interventions have displayed some advantages in reducing feeding issues only when children showed allergies/intolerances or GID problems. Regarding sensorial interventions, since feeding behaviors are influenced by sensorial reactions displayed by children with ASD, displayed as sensibility to taste/smell, temperature, colors, sounds, and movements, these clinical models have shown an advantage in decreasing the related food selectivity. As a consequence, we have listed some interventions based on the desensitization of hyper or hypo stimulations shown by children with ASD. Children with ASD following these interventions showed an increase in no preferred food intake. Finally, more studied behavioral interventions have been described in different sections, showing advantages and disadvantages of each singular or associated behavioral strategy. Lastly, an important section of the current paper is dedicated to parent inclusion and parent training approaches. Commonly, a parent training for parents of children displaying restricted food intake should be planned before and after clinical intervention. Similarly, the clinical staff could respectively collect two data sheets, either for the procedural integrity of the treatment (competencies showed by the parents) or for the child’s progress (e.g., frequency or weight of the accepted foods). Henceforth, in the case of feeding problems during mealtimes, the inclusion of parents would seem to be necessary.

We conclude the discussion by highlighting two recent studies concerning feeding issues as the pandemic of coronavirus 2019 and insight from innovative technologies. Firstly, since research suggests increased adverse behavioral outcomes, such as distractibility and hyperactivity among children with ASD as a result of coronavirus 2019 (COVID-19), little is known about how the pandemic has impacted food-related behaviors. A study has reported the impact of the pandemic on access to preferred foods and eating behaviors among children with ASD [140]. Caregivers (n = 200) participated in a cross-sectional online survey investigating the impact of COVID-19 on reported food and eating behaviors of children, ages 2–17 years. A majority of respondents reported a moderate-to-large impact on their child’s eating behaviors (57%) since the onset of COVID-19, and 65% reported the unavailability of their child’s preferred foods. Reported frequencies of a consumption of meat, seafood, vegetables, and 100% fruit juice significantly decreased among the children post-onset of COVID-19, while the frequency of consumption of sweets increased. A large proportion of caregivers reported substantial COVID-19 impacts on food availability and eating behaviors of children with ASD, especially among low-resource dyads. This study highlights the added burden of existing disparities due to the pandemic on children living with ASD. Furthermore, the use of technological intervention has proven to be an effective tool for educating children with ASD in mastering new skills as compared to traditional methods [141]. Some of the popular technologies are computer-based interventions and robotics, which do not support ecological validity. Consideration of natural factors is essential for better learning outcomes and generalized skills and which can easily be incorporated into reality-based technologies such as virtual reality, augmented reality, and mixed reality. These technologies provide evidence-based support for the ecological validation of intervention and sustaining the attention of children with ASD. A recent review [142] has reported reality-based technology intervention for children with ASD, recommending what technology can support the ecological validity of food intake intervention. Concluding, innovative technologies currently represent an effective tool for teaching children with and without learning disabilities. At the same time, we have reported studies regarding the use of video modeling, a well-established teaching strategy. Recently, virtual environments have allowed us to manipulate ethically uncountable stimuli into different scenarios, offering a child a comfortable virtual reality to address food reticence. On the other hand, practitioners, along with parents, can receive low-cost training to support their children. Finally, future studies could explore the desensitization of sensorial aspects of the foods, as well as the efficacy of the behavioral interventions, including parent training.

## 6. Conclusions

Concluding, food selectivity is a problem common in many children, in particular those with autism symptoms. Therefore, families along with clinical staff should establish concurrent causes of feeding issues in their child through a correct assessment. Successively, the rehabilitation equips the family with the opportunity to choose an appropriate intervention among those who have shown clinical evidence. Currently, despite the complexity of food selectivity in children, we have different clinical instruments to address the phenomenon by providing support to their families through various effective interventions (nutritional, medical, sensorial, behavioral, and environmental). The current narrative review offers a global clinical and educational protocol to people addressing food selectivity in children, especially with ASD. Finally, many of the approaches indicated either regarding the assessment or intervention could assist also children with or without other neuropsychiatric issues.

## Figures and Tables

**Table 2 ijerph-20-05092-t002:** Representative items of the educational caregiver feeding practices.

Parents Respond to Similar Behaviors Providing the Occurrence of These Situations	
Putting the child on the chair to let him eat	Promising the child something as a reward (no food)
Asking questions to the child about the dish	Encouraging the child to eat by preparing food (drawing smiley on the omelet)
Reasoning with the child to get him to eat (milk is good for your health!)	Complimenting the child when he eats (good boy!)
Reproaching the child though does not eat dinner	Inviting the child to eat dinner (your dinner is getting cold!)
Allowing the child to choose the dishes he wants to eat	Punishing the child (if you don’t finish the meat, you don’t play after dinner).
Punishing the child (if you don’t finish the vegetables, you won’t eat the fruit)	Saying something positive about the dish the child is on eating during dinner
Feeding the baby (physically)	Help the child eat dinner (cutting food into small pieces)

## Data Availability

Not applicable.

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
