# Peer review of "Food Selectivity in Children with Autism: Guidelines for Assessment and Clinical Interventions"

_ijerph, 2023, doi:10.3390/ijerph20065092_

Round 1

Reviewer 1 Report

In this review, the authors discussed feeding problems in ASD children, including dysfunctional behaviors at mealtimes. They proposed possible explanations of food selectivity in these children and provided guidelines to clinicians for both assessment and clinical interventions. Moreover, they indicated effective recommendations for parents.

In my opinion, this is an exhaustive, well written and structured review. I only have minor points that could be useful to improve the manuscript before publication in International Journal of Environmental Research and Public Health.

      - An outline of the paragraphs is needed for an immediate understanding of the topics investigated in the review.

-     - Paragraph “1. Introduction” is too long and should be summarized. Moreover, the authors reported: “Likewise, we have listed some of the research-based studies on MEDLINE of the last 10 years, where it is possible discovering inconstant results”. However the methods used for this search should be better explained.

 -   - The same typos occur in the manuscript (i.e. me-ta; is-sues; in-creasing) and they should be corrected.

     - The contribution of the gut microbiota in food issues in ASD children should be better emphasized in the Discussion, also according to recent literature data. Can the analysis of the gut microbiota composition be helpful in the assessment and in driving clinical interventions?

Author Response

Thank you very much for having to revise our manuscript and adding useful suggestions.

  1. As requested, we have added an outline describing the various paragraphs of the paper.
  2. We ha summarized the introduction as suggested.
  3. We have disambiguated the paragraph concerning the nutritional inadequacies and propose a new logical structure for this section Diagnosis->impact on nutritional requirements, associated GID problems and weight/diet->theories explaining the phenomenon->parenting stress.
  4. As you can note, we have modified the substantiality correctness, accuracy, grammar, syntax, and formality throughout the paper (in each section).
  5. As you suggested, we have added to the discussion a contribution of the gut microbiota in the explanation of the multidimensional model, combining it with the previous paragraph on GID issues.

Once again, thank you very much for your support through which we have meliorated the quality of the paper. Let me know if you have new suggestions for the manuscript.

Sincerely yours.

XY 

Reviewer 2 Report

1. Please proofread and correct the typos in the manuscript. There are some unnecessary hyphens. 

2. Please add a level 1 heading to chapter 1 to separate it from the Introduction. 

3. Please be focused on the topics of each heading when assembling the contents. For example, chapter 1.1 is discussing food selectivity in children with ASD. In my opinion, the relationship of overweight with psychiatric disorders has nothing to do with this topic, so can be removed. Similar issues are also found in the other chapters. Please proofread and re-edit the manuscript. The current version has too many tedious, unrelated details. 

Author Response

Thank you very much for having to revise our manuscript and adding useful suggestions.

  1. As you can note, we have modified the substantiality of the correctness, simplicity accuracy, grammar, syntax, and formality throughout the paper (in each section).
  2. We have changed the previous structure of the first paragraphs as suggested, summarizing the introduction, and adding an outline before the first section.
  3. Moreover, we have changed the order of the first paragraphs in order to increase the readability of the paper.
  4. As suggested, we have eliminated diverse parts throughout the manuscript, substantially the new version of the paper is really different since we have canceled redundant parts and no useful details. Also, the part concerning the association between overweight and psychiatric disorders has faded where necessary.

Once again, thank you very much for your support through which we have meliorated the quality of the paper. Let me know if you have new suggestions for the manuscript.

Sincerely yours.

XY 

Round 2

Reviewer 2 Report

The revised version addressed the major concerns in the first manuscript. I accept the present form for publication.